# Vestibular Stimulation May Drive Multisensory Processing: Principles for Targeted Sensorimotor Therapy (TSMT)

**DOI:** 10.3390/brainsci11081111

**Published:** 2021-08-23

**Authors:** Brigitta Tele-Heri, Karoly Dobos, Szilvia Harsanyi, Judit Palinkas, Fanni Fenyosi, Rudolf Gesztelyi, Csaba E. More, Judit Zsuga

**Affiliations:** 1Department of Health Systems Management and Quality Management for Health Care, Faculty of Public Health, University of Debrecen, Nagyerdei krt. 98, 4032 Debrecen, Hungary; brigittaheri@gmail.com (B.T.-H.); doboskaroly@gmail.com (K.D.); harsanyi.szilvia@sph.unideb.hu (S.H.); palinkas.judit@sph.unideb.hu (J.P.); 2Kálmán Laki Doctoral School, University of Debrecen, Nagyerdei krt. 98, 4032 Debrecen, Hungary; 3BHRG Foundation, Tavasz u. 5, 1033 Budapest, Hungary; fenyosi.fanni.mentor@gmail.com; 4Department of Pharmacology and Pharmacotherapy, Faculty of Medicine, University of Debrecen, Nagyerdei krt. 98, 4032 Debrecen, Hungary; gesztelyi.rudolf@pharm.unideb.hu; 5Department of Psychiatry, Faculty of Medicine, University of Debrecen, Nagyerdei krt. 98, 4032 Debrecen, Hungary; more.e.csaba@med.unideb.hu

**Keywords:** sensorimotor integration, multisensory integration, vestibular stimulation, TSMT

## Abstract

At birth, the vestibular system is fully mature, whilst higher order sensory processing is yet to develop in the full-term neonate. The current paper lays out a theoretical framework to account for the role vestibular stimulation may have driving multisensory and sensorimotor integration. Accordingly, vestibular stimulation, by activating the parieto-insular vestibular cortex, and/or the posterior parietal cortex may provide the cortical input for multisensory neurons in the superior colliculus that is needed for multisensory processing. Furthermore, we propose that motor development, by inducing change of reference frames, may shape the receptive field of multisensory neurons. This, by leading to lack of spatial contingency between formally contingent stimuli, may cause degradation of prior motor responses. Additionally, we offer a testable hypothesis explaining the beneficial effect of sensory integration therapies regarding attentional processes. Key concepts of a sensorimotor integration therapy (e.g., targeted sensorimotor therapy (TSMT)) are also put into a neurological context. TSMT utilizes specific tools and instruments. It is administered in 8-weeks long successive treatment regimens, each gradually increasing vestibular and postural stimulation, so sensory-motor integration is facilitated, and muscle strength is increased. Empirically TSMT is indicated for various diseases. Theoretical foundations of this sensorimotor therapy are discussed.

## 1. Introduction

While sensory organs develop in utero, e.g., the mature neonate is born with a fully functional vestibular organ, higher order sensory processing is yet to mature in the full-term neonate [1]. Multisensory processing, a fundamental mechanism for disambiguating complex environmental signals [2], if impaired may be implicated in a multitude of diseases and disorders. Previous reports have established compromised multisensory integration (MSI) in delayed motor development [1], moderately severe to severe cerebral palsy [3,4], intellectual disability [5,6], autism spectrum disorder [7,8], problems with attention including diagnosed attention deficit, hyperactivity disorder [9], sensory organ dysfunction [10] and presence of sensory processing disorders [11]. Furthermore, as MSI seems necessary for language development [12,13] its disruption leads to learning disabilities e.g., dyslexia [14].

The current paper lays out a theoretical framework to account for the role vestibular stimulation may have driving multisensory and sensorimotor integration. This framework offers a testable hypothesis explaining the beneficial effect of sensory integration therapies for attentional processes. Furthermore, implications of these processes regarding a form of sensorimotor therapy are discussed.

The vestibular system is fully operational in the term infant and offers continuous gravitational sensory input that will affect the whole brain [15]. Vestibular sensory perception manifests as a ubiquitous model of gravity in the human brain, which evolves through the interaction with the environment. A fundamental attribute of the vestibular system is compilation of the absolute geocentric, idiothetic reference frame by determining absolute body motion in gravitational space. Input from visual and somatosensory modalities on the other hand contribute to the development of egocentric and allocentric reference frames, respectively [16]. This allows distinguishing movement in the environment from self-movement. Establishing an idiothetic coordinate system encompassing representations of verticality will enable differentiation of spatial relations in a three-dimensional space [16,17]. This will form the basis of postural and motor coordination, fine motor control and visual processing [18,19]. As such, vestibular signals may be used to disambiguate conflicting or inaccurate information by reconciling diverse signals [20,21]. Hence vestibular function forms the basis of postural and motor coordination, as well as gaze stabilization [17,22]. Both linear and rotational input will work to stabilize the head on neck and body, stabilize gaze during active and passive head and body movement [17]. In fact, the vestibular system is instrumental in coordinating head movement with eye movement, offering the foundations for postural tonic control and coordinated control of the eyes [20,23]. It is the vestibular system via the vestibulo-ocular reflex (VOR) that compensates for head rotations above 100°/second, a velocity often encountered during usual daily activities, allowing for stabilization of foveal gaze on target [22]. Feedback received regarding the timely and coordinated visual, proprioceptive and vestibular input will constitute the formation of spatial representation both of self and the environment [17]. This constant vestibular backdrop will be used to refine the rudiments of inborn body schema [24] by decoding the multisensory input offered by primitive reflexes [16], and related spatial cognition [16]. Spatial representations are mandatory for higher level cognitive processes such as spatial memory, mental calculations, object-based mental transformations, social cognition and emotional regulation [16,17]. Furthermore, vestibular signals underlie change of perspective from an egocentric to an allocentric viewpoint, by means of visuo-spatial manipulations. This enables taking third-person perspective mandatory for empathy, understanding and predicting emotions and intentions of others [25]. Spatial memory enables internal simulation and re-representation of the sensory-motor loop’s activity in anticipation of future events [17], contributing to a cognitive map possibly used by other processes e.g., model-based reinforcement learning (for an overview see [26,27]). Conversely patients with vestibular dysfunction have been described to suffer from short-term memory loss, concentration, impaired VOR leading to reading disabilities, impaired ability to estimate basic numeric attributes of the environment, such as distances and weights, translating into poor arithmetic skills [16]. Vestibular impairment, furthermore, can impair self-perception and body schema construction. This hinders awareness of body parts, distinction of self from the environment resulting in feelings of depersonalization and out of body projections [16,17].

## 2. Materials and Methods

The aim of the study was to present an overview of core neurological processes that drive maturation and development of the central nervous system. The theoretical framework laid out provides a novel mechanism explaining how vestibular stimulation may facilitate MSI in the SC. The merit of this concept is that it gives rise the several testable hypotheses carrying clinical and pedagogical implications. The current work builds on former experimental and clinical work of our group and others cited throughout the text, allowing the formation of this theory.

## 3. Results

### Development of Multisensory Integration

MSI is a process for combining sensory information obtained from distinct sensory channels on the level of the single neuron. Albeit ubiquitous, with a long evolutionary history, it is not coded in the genome. Its maturation shows a protracted course during postnatal life [28,29,30]. Multisensory maturation is driven by experience that enables linking cross-modal attributes of self-referential and environmental events to each other. Hence sensory experiences form the basis for developing a world model based on perceived cross-modal contingencies [28]. The vestibular system offers a prime example for resolving sensory ambiguity by means of MSI yielding a novel sensory representation that serves to differentiate between voluntary actions and environmental perturbations [31]. Conversely, MSI based on integration of sensory information weighted according to the precision of sensory cues was shown to underlie optimal motion and orientation estimates when sensorimotor integration was investigated using a limb movement control task [32]. Corroborating evidence was further provided regarding sensorimotor postural control where dynamic reweighting of sensory and orientation cues explained behavior [33].

The brain is adept to function in noisy environments, where sensory signals vary in terms of their intensity, relative timing and spatial location as purposeful behavior must persist despite uncertainties and unexpected perturbations [2]. One approach for the brain to cope with the multitude of signals, is abstracting distinct yet related sensory attributes by dedicated multisensory neurons [20]. By combining independent estimates of unimodal sensory input, noise may be reduced in parallel with increased precision, as complementary or redundant signals are generally optimally integrated (for a review see [1]). Computational theories depicting the combination of unisensory attributes of sensory cues for multisensory processing utilize Bayesian decision theory models [34,35,36]. This posits that the brain decodes hidden environmental regularities by means of statistical inference, and as such computes the probability distributions of prior knowledge and estimates the likelihood of sensory information [2]. Multisensory processing is consistent with Bayesian with principles that describe perception rooted in MSI and sensorimotor behavior [37].

Moreover, considering MSI under the premises of Bayesian decision theory models, perception may be regarded as inference about the causes of sensory cues, and attention may be formalized as the precision of environmental and self-referential sensory signals [28,38]. Hence attention manifests as weighting data with respect to estimated precision [38,39]. Furthermore, assuming that the only unexplained sensory details of the environment are prediction errors, attention derived weights reflect estimated precision, biologically implemented as synaptic gain [38,39]. Hence multisensory processing serves to optimize precision-weighted prediction errors. This concept explains how multisensory experience enhances the salience of events if stimuli fulfill certain conditions for temporal synchrony, spatial congruence and stimulus efficacy [2,40,41]. Corroborating evidence shows correlated activity of unisensory afferents of the anterior ectosylvian sulcus (AES) and their respective multisensory postsynaptic common target neurons in the superior colliculus (SC) as the results shows how synaptic weights are adjusted in line with Hebbian learning rules [28]. Summarizing, it would follow that intersensory redundancy derived from concurrent presentation of the same information over several modalities, by increasing precision of the multimodal percept, would lead to heightened attention.

Given that MSI serves to craft a model of the environment that is specific for the individual’s cross-modal experiences, MSI develops if different sensory signals are in close temporal and spatial correspondence, possibly indicating their probable joint origin from common events [28]. Conversely, MSI is more pronounced if the relationship between stimuli is in line with prior expectations or congruent with natural interrelation of senses [29]. Temporal congruence of two stimuli mandates that distinct modalities reach the target multisensory convergence area within a temporal window that can adjust for different architectural delays inherent of said sensory modalities [42]. For example, tactile latencies are on average 40 ms shorter than visual latencies in the posterior parietal cortex (PPC) [40] and visual cues precede auditive ones [43]. Additionally, amodal properties such as temporal synchrony, common rhythm and spectral information between cross-modal stimuli facilitate integration. This process is fundamental for infants acquiring speech, as intrinsic rhythmic regularities and temporal correlation between facial and lip movements and auditory input drives MSI for perceptual processing needed for speech [43,44]. Congruent audiovisual stimuli presenting the same speech input was shown to improve accuracy of recognition when compared to audio or visual stimulus administered alone [45].

Central to the process of integrating multisensory information is the concept of reliability, that is incorporated into computational models as weights that are continuously re-calibrated as behavior adapts to its environment. Multisensory neurons have overlapping receptive fields that are responsive to their unisensory input at birth, but their ability to integrate multisensory input is the result of sensorimotor experience-driven maturation [30]. Accordingly, the ability for MSI is not innate, it needs to be learnt [30,46]. For example, previous work has shown that statistically optimal integration of haptic and visual information for postural control develops gradually, as it is underdeveloped in 8-year-old [47] but approximates adult maturity in 10-year-old children [48]. Concordant to these findings, is the suboptimal integration of cues for navigation in children younger than 8 years of age [47]. Cuturi and Gori [21] have shown that vision and touch are both influenced by the same vestibular/proprioceptive priors regarding head and body orientation with respect to gravity. Furthermore, children as young as 6 years of age showed some level of optimal statistical inference in a simple position-estimation Bayesian task, where data was best fitted by Bayesian model fit. These findings enabled the authors to conclude that some level of Bayesian integration is already present in early childhood, which is gradually refined parallel to acquiring experience, as the brain learns to approximate [37]. Alternatively, cross-sensory calibration may be the antecedent of MSI, allowing the more accurate sense to teach the others. For example, the haptic system may calibrate vision, while the visual system may calibrate hearing for assessing space and hearing seems to calibrate vision with respect to time perception [1].

In addition, the strength of the signal plays an important role for MSI. Inverse effectiveness, a universal phenomenon observed during multisensory enhancement is the ability of weak sensory signals to elicit a superadditive response, if cross-modal stimuli are coincident in time and space and fall on overlapping multisensory receptive fields [2,49]. Increasing signal saliency decreases the potential for multisensory enhancement as unisensory responses are already significantly developed [40]. Hence the ability to detect subthreshold cross-modal stimuli may be explained by the process of inverse effectiveness and superadditivity [50]. For example, temporal regularity and cross-modal temporal cueing, congruency prerequisites for MSI were shown to lower auditory thresholds reflected by improved auditory perception in response to subthreshold auditory responses [43]. Additional to speech perception benefiting from temporal correlation of visual cues in relation to audition, facilitation of perception was further shown with regards to other combinations of subthreshold stimuli including olfactory and visual modalities in Drosophila [29]. Taking the principle of dual coding into consideration, e.g., that if information is processed through multiple senses, limited processing capabilities may be circumvented, it would follow that interventions facilitating multisensory learning may help overcoming difficulties stemming from impairment of one or more sensory organs.

Evidence shows that multisensory neurons are present at birth in various areas of the brain, dedicated to multisensory processing, including SC, the parieto-insular vestibular cortex (PIVC), and PPC [49,51,52]. General concepts of MSI are derived from the canonical model, the cat SC, as this structure has an abundance of multisensory neurons and may be conveniently studied under experimental conditions. Preclinical studies show that although SC neurons are responsive to more than one sensory modality at birth [28,49], they are incapable of synthetizing cross-modal responses, therefore the emergence of novel unitary behavior is absent [49]. This is dependent on ipsilateral corticotectal inference received from the association cortex (the AES and the rostral lateral suprasylvian sulcus in particular) that develops 4 to 12 weeks after birth [52]. That influence from ipsilateral association cortex is a prerequisite for MSI was corroborated by different paradigms. Reversible, chronic 4 to 8 weeks long pharmacological deactivation of the association cortex in early-life, rendered the association cortex insensitive to cross-modal experiences. Delay of timely acquisition of sensory experiences required for SC maturation caused protracted delay in maturation of MSI up to 1.5 to 4 years [53]. Conversely, selective deactivation of this area impaired SC multisensory neurons’ ability to integrate cross-modal inputs underlying MSI [54]. Further indirect evidence regarding the permissive effect of the association cortex with respect to the functional capacity of SC multisensory circuit is the correlated activity elicited by cross-modal cues in the SC multisensory neurons and their unisensory cortical afferents [54]. Hence the association cortex may serve as a portal that allows adaption to specific experience-based multisensory experience via altering the functional capability of SC multisensory circuit [55]. In fact, albeit an environment devoid of early cross-modal challenge hinders SC’s ability to integrate converging sensory inputs, this activity does re-emerge if cross-modal experience is obtained later life [54]. Previous reports showed that dark-reared cats were able to re-establish their capacity for MSI of paired visuo-auditory stimuli in a month after being exposed to weekly visuo-auditory stimulation. This suggests that SC multisensory neurons retain considerable plasticity during later stages of development allowing adaptation [54]. This implicates the possibility for synergistic beneficial effects between sensory integration therapies and corrective measures for sensory organs (e.g., correcting visual acuity and hearing), as the tailored sequences of exercises rich in invariant cross-modal features along with enhanced sensory perception of sensory stimuli could facilitate multisensory learning.

Although the homologue area of cat AEC is not known in primates [49], we point to two candidate multisensory cortical areas that may assume a role similar to the cat AES in terms of MSI. The first one is the primate PIVC+, an area possibly made up of two distinct anatomical and functional subareas, the posterior insular cortex (PIC) and PIVC [51]. PIC is a visuo-vestibular area, while PIVC is a multisensory cortical region with neurons responsive to vestibular (angular acceleration), somatosensory (e.g., stimulation of neck and shoulder mechanoceptors while the head is stable) and visual (especially optokinetic) stimuli [51,56]. PIVC is considered a hub for vestibular cortex and there is some evidence for a human homologue area as well [57,58]. The second one is the PPC responsible for transforming sensory signals into reference frames that participate in guiding gaze or reach [49]. Some parts (e.g., the ventral intraparietal area) were shown to receive direct vestibular projections, hence they are considered to be part of the vestibular cortical network [59]. Furthermore, using horseradish peroxidase retrograde labeling, corticotectal cells were identified in both the parieto-insular vestibular cortex and the posterior parietal cortex [60]. These findings together tempt the speculative notion vestibular stimulation may enhance maturation of PIVC and the PPC (Figure 1). Consequently, PIVC and/or PPC will via their corticotectal afferents possibly facilitate MSI in the SC. This putative mechanism may be compromised in preterm infants constrained to incubators and those needing prolonged hospitalization after birth, thus explaining later, seemingly, unexplained developmental delays [61,62,63]. The findings of Weinstein and colleagues [64] showing deviant sensory-motor reactivity in preterm children paralleled by disturbed integrity and lagging maturational level of early and late maturing fiber tracts further support this notion, given the significance of early experiences and need for intact corticotectal projections [28].

Benefits of MSI are relevant if they manifest in behavioral outputs such as superadditivity or subadditivity, a nonlinear combination of modality-specific influences. Pronounced enhancement exceeding the sum of individual neuronal responses may ensue under conditions of MSI if two unimodal stimuli fall within overlapping excitatory receptive fields. Under these circumstances spatial congruence of stimuli is fundamental to allow co-activation in multisensory areas by unisensory modalities. During development this evolves by sculpting of spatially overlapping receptive fields from their larger neonatal templates, based on early experiences [28]. This process is substantial from infancy as compiling sensory experiences from birth will underlie the brain’s ability to model the world using Bayesian inference. Conversely, sensory representations of multisensory areas e.g., the SC, the PPC, are topographically organized and show spatial correspondence since multisensory neurons have multiple excitatory receptive field, one for each modality they respond to [49]. For example, the receptive field for visual and auditory modalities of an audiovisual neuron would correspond, if both the visual and auditory receptive fields would be for example in the nasal space [52]. Hence a cross-modal activation with respect to a single event will repeatedly activate the same multisensory neuron through separate sensory systems. Many excitatory receptive fields are however bordered by an inhibitory region. Therefore, if spatial location of cues becomes distanced so activation from a modality falls on this inhibitory border zone it will inhibit the activating effect of the other modality, causing depressed neuronal response, and consequently degrades performance [49]. Furthermore, if one of the sensory cues is outside of the receptive field, the integrated response will be absent [52]. Following the spatial principle of MSI (see above) while maintaining independent movement of each sensory organ, modality-specific receptive fields may be linked to eye position a phenomenon described in the SC and the PPC [65,66,67]. Hence, this oculocentric coordinate frame enables adjustment of visual, auditory, somatosensory or cross-modal goals with respect to the orientation of gaze, yielding a unified coherent locus of activity within the sensory-motor map [49]. However, intermediate reference frames, receptive fields that shift only partially in response to the position of the eyes were also described previously [49]. These were implicated to enable transformation between distinct coordinate frames, for example from eye to head or to body centered frames [68,69]. Transitioning from a head centered to a body centered reference frame is mandatory to interpret the relationship between body motion and vestibular stimulation. For example, PIVC, the multisensory vestibular cortical area, was shown to have a reference frame that is intermediate between head and body centered and showed weak modulation by eye position [70]. Similarly, intermediate reference frames for head and body centered movement were reported in the cerebellum [20] in an area containing bimodal neurons that integrate vestibular and cervical proprioceptive inputs. Brooks and colleagues have further shown that in adult monkeys with intact vestibulospinal reflexes, proprioceptive and vestibular signals cancel each other out when the head is passively moved relative to a stationary body [71]. Misalignment of receptive fields, thus, may hinder cancellation of different signals, possibly allowing for a maladaptive motoric response.

Observations further depict that neonatal MSI develops more rapidly if multisensory stimuli are presented with invariant cross-modal features, showing limited ambiguity in a controlled setting, as opposed to the general rearing conditions where relative signal intensities, timing and spatial alignment as well as competing cues cause considerable ambiguity [54]. It was also shown that dark-reared cats were able to re-establish their capacity for MSI of paired visuo-auditory stimuli in a month after being exposed to weekly visuo-auditory stimulation, suggesting that SC multisensory neurons retain considerable plasticity during later stages of development allowing adaptation [54].

## 4. Discussion

Targeted sensorimotor therapy (TSMT) is a form of sensorimotor integration therapy developed in Hungary. Twenty years have passed since its initial introduction, and a formal further education program was developed for teachers, special needs educators, physiotherapists and physicians to become an accredited TSMT therapist. Each module of the educational pathway is accredited by the Hungarian Educational Authority [72] and upon completion the candidate becomes a certified TSMT therapist. Albeit over the 2 years 100 accredited therapists managed 12,532 children [73] only limited evidence is available regarding the its therapeutic efficacy. Empirically, TSMT may be indicated for varying diseases and disorders including delayed motor development, delayed language development, moderately severe to severe cerebral palsy, mental retardation, problems with attention including diagnosed attention deficit, hyperactivity disorder, sensory organ dysfunction or presence of sensory processing disorders. Children having learning difficulties, high risk for dyslexia, dysgraphia, dyscalculia as well as those with behavioral and social difficulties including impulsivity, poor task compliance, aggression, lack of cooperation or autism spectrum disorder may also benefit [74].

A previously published small, prospective, single-center case series reported the effect of a one-year long TSMT therapy course [75]. Children were referred by the Department of Neurosis of the Children’s Hospital of Buda, by educational guidance services and by outpatient child neurology services, amounting to the referral of 24, 5 and 3 children, respectively. Children were prospectively recruited to participate in the study, given they were diagnosed with attention-deficit/hyperactivity disorder, were aged between 4 years 0 days and 5 years 0 days, and the parents gave consent to participate in the year long program. Children received 90 min long TSMT therapy sessions twice a week, for one year, in the outpatient unit of the BHRG Foundation. As TSMT sessions were delivered by a single therapist, who had over twenty years of experience in the area of child development, inter-therapist variation was omitted. Children were assessed by the Condition and Movement Test (CMT) at baseline and during follow-up. This test is designed to measure neurophysiological maturity [74,76,77]. It assesses five domains including primitive reflexes, motoric-control, body schema perception/spatial orientation/dexterity, tactile sensory system and rhythm. Scores are expressed as a percentage, with lower scores reflecting impairment, with values lower than 50% indicating significant development delays, whereas values exceeding 75% reflecting typical neurodevelopment and maturity. Moreover, higher scores were shown to predict better school performance reflected by better grades [78]. At 12-month follow-up the children were 6 years old and received a rigorous one-year regimen of TSMT therapy. Overall, data of 25 children were included in the analysis as 6 children were lost during follow-up. Measures of central nervous system maturity showed significant improvement reflected by changes in scores for primitive reflex profile (mean at baseline 21.42 ± 8.74% vs. mean at follow-up 50.71 ± 12.41% *p* < 0.001), motoric-control (mean at baseline 17.5 ± 9.02% vs. mean at follow-up 45.75 ± 13.10% *p* < 0.001), body schema, spatial orientation and dexterity (mean at baseline 39.23 ± 8.10% vs. mean at follow-up 77.23 ± 11.76% *p* < 0.001), tactile sensory system (mean at baseline 42.5 ± 12.31% vs. mean at follow-up 68.21 ± 7.34% *p* < 0.001) and rhythm (mean at baseline 7.2 ± 8.42% vs. mean at follow-up 35.6 ± 8.69% *p* < 0.001). Furthermore, results have shown that after one year of TSMT therapy several domains of neurodevelopment (e.g., motoric control, body schema perception and tactility) became comparable to that observed in typically developing 6-year-old preschoolers [74]. A successive case report has also described the beneficial effect of TSMT in a 30-month-old boy who showed significant developmental delay (delayed movement and language development, lack of voluntary bowel/bladder control, sleep problems). Following 26 weeks of TSMT therapy the child’s development significantly improved, he began speaking a few words (mom, mamma, papa and other imitative words), and domains of self-regulation also showed improvement [75].

The goal of TSMT is task completion in the context of delivering stereotyped sequences of exercises rich in invariant cross-modal features. We hypothesize that these exercises by facilitating MSI and sensorimotor learning contribute to developing acoustic, vestibular, proprioceptive, tactile and visual modalities, adaptive motor responses and attention. We propose that the efficacy of TSMT comes from strong vestibular stimulation, as activation of the parieto-insular vestibular cortex, and/or the posterior parietal cortex TSMT therapy may provide the cortical input for multisensory neurons in the superior colliculus that is needed for multisensory processing. Furthermore, we suggest that intersensory redundancy derived from concurrent presentation of the same information over several modalities, may be the reason for improving attention by increasing precision of the multimodal percept.

TSMT is based on several fundamental principles. It is a regressive therapy as it strongly builds on the most archaic system, the vestibular system, to facilitate ontogeny of adaptive behavior. A further fundamental attribute is the exploitation of rhythm, audio-visual-motor synchronization, and vestibulo-ocular reflex. The training program is targeted, as treatment regimens are individualized based on the findings of a detailed physical examination (e.g., CMT, described above). This includes a rudimentary assessment of vision, hearing, muscle tone and hearing allowing early diagnosis and indication of corrective measures (hearing aids, glasses, etc.). This is supplemented by assessing the child’s current and past case history and the environment.

The recommended age for TSMT is between 6 months to 12 years of age. TSMT utilizes specific sensory tools (colorful beanbags, balls, etc.) and instruments, e.g., large fitness balls, skateboards, rotating chair and specialized tilting board. The later are crafted to aid the administration of vestibular stimulation via both the semicircular labyrinth and the otoliths and facilitate MSI underlying development of adaptive postural control.

TSMT is delivered using a primary therapist model. Treatment courses are administered in successive 8-weeks long intensive treatment periods (a sample treatment regimen is illustrated in Figure 2) with no-treatment periods interspersed between three eight week-long sessions. Each regimen builds on the preceding one, gradually increasing vestibular and postural stimulation. The sequence of exercises is determined by the therapist on a case-by-case basis. As vestibular stimulation is a fundamental component of TSMT regimens, usually the first therapeutic regimen contains the highest proportion of passive vestibular exercises. These are generally administered at the beginning of the therapeutic sessions as vestibular stimulation is known to induce arousal via direct projections of the vestibular nuclei to locus ceruleus [79]. Passive vestibular exercises include tasks in which the head’s position relative to the body is stationary, rotated to either side, flexed or extended. Active vestibular stimulation includes tumbling, and active exercises on the skateboard. Linear (translational) and rotational acceleration are also delivered using different sensory-motor equipment (balls, skateboard, tilting board, etc.). Strong vestibular stimuli are administered by exercises in an upside-down position, tumbling and swinging. Different static postures (e.g., supine, prone, hands and knees, sitting, erect), automatic (e.g., rolling, quadrupedal locomotion—creeping and crawling) and goal directed motoric patterns (e.g., spider, crab and giant walking and bear crawling) are achieved during passive and active exercises.

As TSMT is fundamentally a sensory-motor integration therapy, motor planning of voluntary movement is also challenged. Exercises for ipsilateral or contralateral synchronization of extremities, cross-patterns and serial movement patterns consisting of two to eight sequential movements are also fundamental in TSMT. Development of repetitive movement patterns is usually accompanied by nursery rhymes chanted by both the therapist (parent) and possibly the child, in a way that chanting is aligned to the rhythm of the exercises. In fact, the motto of TSMT is that if a therapy is silent, it is not TSMT. Hence TSMT utilizes co-administration of stimuli that share congruent amodal properties. Verbal instructions are given in synchrony with the rhythm of a motor task for example when bouncing on the fitness ball, using the balance board, jumping, quadrupedal locomotion, etc., to facilitate verbal-motor synchronization. During therapy, the child has the ability to observe the face of the therapist (parent) while talking to allow accumulation of experience mandatory for speech acquisition. Instructions are given so the therapist faces the child and if possible, maintains eye contact while assisting with the exercises. This facilitates understanding by allowing visual access to the face, also assists receptive language development.

Efficacy of TSMT may be interpreted within the frame of dynamic systems theory (DST), that offered a fundamentally new theoretical approach to neurodevelopment. Traditional views regarding motor development aim to achieve functional improvements by developing “typical” patterns of movement, thus placing the “correction” of the child’s body function into the focus of therapy [80]. On the contrary, DST offers a context-based approach, by proposing that behavior emerges as individual characteristics, environment and task constrains converge in a coordinated manner [81]. Since behavior is viewed as the net of the body-environment-task interaction, developmental trends appear due to the changes of interaction between the components of the system. Therefore, causal factors in development are not important, instead DST focuses on processes influencing the components of emerging behavior [82]. Hence therapeutic interventions influenced by DST center on changing the constrains of the environment and task; to attain improved functional performance and consequent motor development [83]. Furthermore, dynamic systems theory places great emphasis on spontaneous self-organization, a stable, preferable movement pattern based on the influence of constraints, with task or activity completion as the goal (as opposed to learning the typical movement patterns). By changing the landscape of constraints, e.g., placing the child into different environment or different tasks, and enabling atypical movement with the aim of task completion new, adaptive movement patterns may be learnt [81], leading to multi-sensorial mapping with error signals that subsequently contribute to more and more effective engrams/attractor wells. TSMT incorporates multi-sensory inputs (with weighting of signaling being practiced and refined) into therapy, supplying fundamental input to develop the repertoire of feedback and feedforward. It is interesting to note that this mechanism may be observed in the play of children in many cultures.

## 5. Conclusions

The current article overviews relevant neuroscientific findings and offers novel insight into mechanisms of MSI. We propose that vestibular stimulation, by activating the parieto-insular vestibular cortex, and/or the posterior parietal cortex may provide the cortical input for multisensory neurons in the superior colliculus that is needed for multisensory processing. These mechanisms may be instrumental in sensorimotor integration therapies, nonetheless larger clinical studies are needed.

## Figures and Tables

**Figure 1 brainsci-11-01111-f001:**
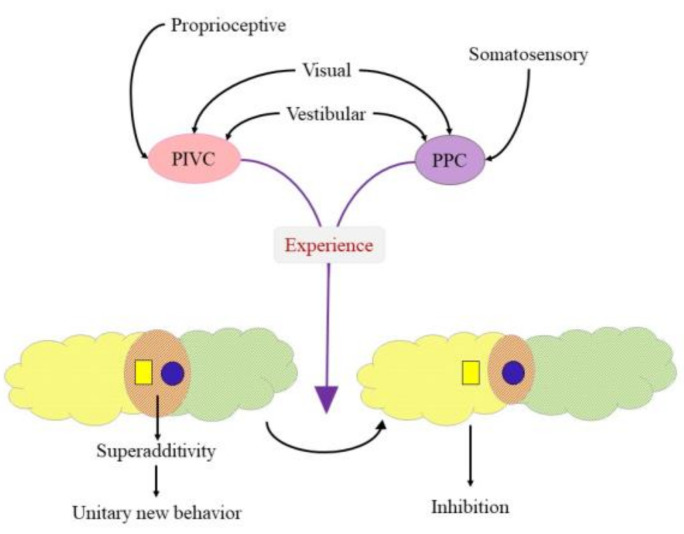
Putative mechanism for vestibular stimuli-driven maturation of MSI. Sensory stimulation activates higher level multisensory vestibular centers (PIVC and PPC). This on one hand provides sufficient input for maturation of multisensory neurons in the superior colliculus. On the other stimulation of the PPC allows change of reference frames and concurrent alteration of receptive fields of SC multisensory neurons. This experience related change may modify the response given to sensory stimuli from superadditive unitary output (illustrated on the left side) to inhibition (illustrated on the right side). PIVC: parieto-insular vestibular cortex; PPC: posterior parietal cortex.

**Figure 2 brainsci-11-01111-f002:**
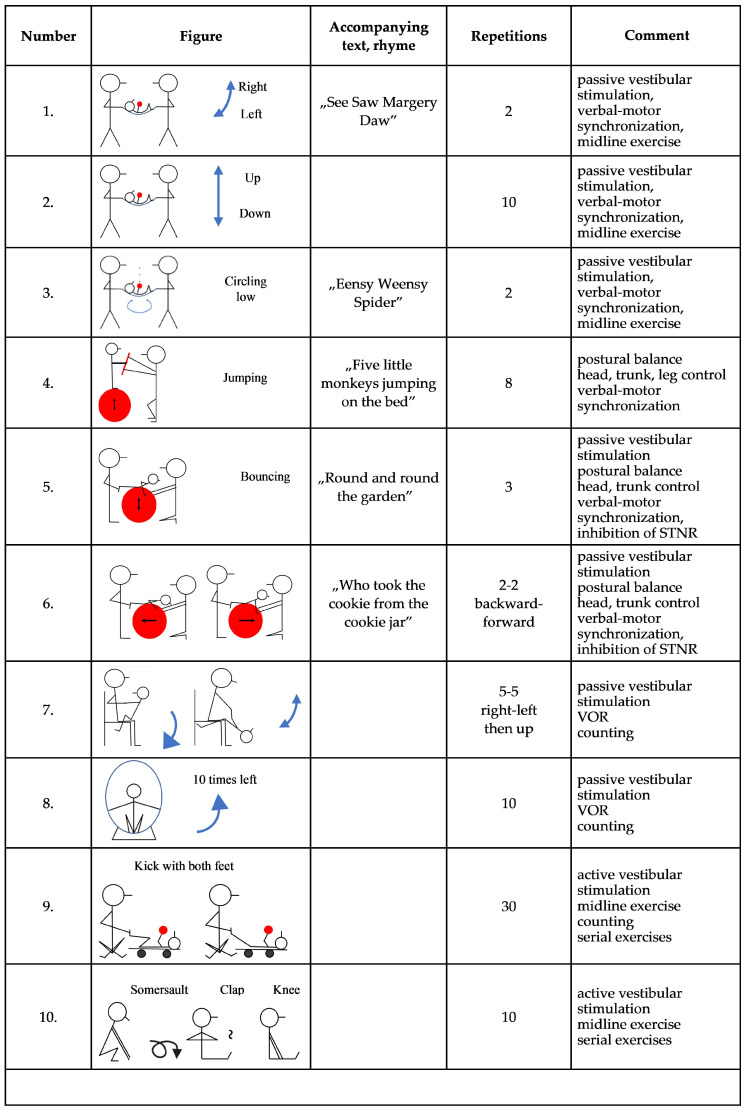
Illustration of an individualized TSMT regimen.

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
