# Peer review of "Vestibular Stimulation May Drive Multisensory Processing: Principles for Targeted Sensorimotor Therapy (TSMT)"

_brainsci, 2021, doi:10.3390/brainsci11081111_

Round 1
Reviewer 1 Report
This review represents very useful and interesting contribution to the literature. The authors mentioned that the TSMT method works as a sensory-motor integration therapy based on sequential repetitive movements patterns. Do the authors have their own speculative hypothesis how these repetitive patterns could influence the integration of the primitive reflexes via inhibitory mechanisms? If yes it would be very useful to mention this in the discussion section.
Author Response
Response to Reviewer 1.
The authors are grateful for your work to review our manuscript. The notion put forward by the manuscript concerning repetitive patterns that lead to the integration of primitive reflexes is highly speculative and currently lacks experimental proof. Due to this, as well as to the robust comments provided by the Academic Editor and Reviewer 2, the aspect of primitive reflex integration in TSMT was omitted from the present manuscript. Nevertheless, as we find this approach useful (or, at least, interesting), this concept may be addressed in a next work of our team.
Thank You again for reviewing our manuscript. We hope that you will find the revised manuscript suitable for publication in the Brain Sciences.
Reviewer 2 Report
Thank you for the opportunity to read this paper. The substance of the paper is essentially a scholarly review of aspects of multi-sensory convergence and integration and its development. Much of the material is based on selected aspects from lower order species. Having said this, those aspects read satisfactorily and whilst not novel are not contentious.
What is contentious is the insertion throughout of the impact of primitive reflexes and their "inhibition" during neuromaturation and the development of motor and postural control. The influence of these so-called primitive survival reflexes has by and large been abandoned in current neuroscience and this paper makes no new claims or evidence to warrant resurrecting them. Hence the paper has a fatal conceptual flaw which then is used to base a justification for a particular therapy.
In turn this therapy is not well evidenced in the specific form presented (one trial n=25) which makes large claims of effectiveness but we are not even sure of the trial methodology. This is simply not acceptable. And even if we were presented with more substantial effectiveness data (and quality of the study provided), the linking of the various exercises to an effect on the putative primitive reflexes has no basis. The effectiveness could just as easily be based on the current dynamic actions/systems thinking around neurodevelopmental processes in terms of multi-sensorial mapping with error signals leading to more and more effective engrams /attractor wells. So the incorporation of multi-sensory inputs (with weighting of signalling being practised and refined) into any therapy is not contentious, it is fundamental to develop the repertoire of feedback and feedforward - it is present in any culture in the play of children (electronic game play is probably the notable exception).
Author Response
Response to Reviewer 2.
The authors would like to thank the conceptual input of Reviewer 2 regarding our manuscript brainsci-1250905. After carefully reviewing the reflections provided, the authors realized that linking primitive reflex inhibition with TSMT was highly speculative, and there was a lack of connection between the various exercises and an effect on primitive reflexes. Due to this realization, the impact of primitive reflexes and their “inhibition” during neuromaturation was removed from the present manuscript, shifting the focus to the role of vestibular stimulation in multisensory integration. The authors would also like to thank the Reviewer 2 for steering our attention toward the dynamic systems theory, a concept relevant for the present manuscript. Owing to this input, a section regarding the possible explanation of effectiveness of TSMT in the context of dynamic systems theory was included in the manuscript.
Overall, thank You again for reviewing our work. We hope that you will find our revised manuscript suitable for publication in the Brain Sciences.
Round 2
Reviewer 2 Report
I thank the authors for attending to the issue of primitive reflexes throughout the entire paper.
I still find little justification in linking their initial scholarly work to their sensori-integration approach. However if the editors feel this is OK to have a non-critical appraisal then so be it.
However the recent inclusion of the Ayres SIT work is not necessary - this approach is very poorly evidenced internationally and should be removed.
Further i appreciate the inclusion of the DST summary. However certainly the main criticism of SIT (apart from the lack of evidence for effectiveness) is that it runs counter to DST. It is based on rejected neurophysiology. Somehow the authors havent picked this up?
